# The Competitions, Negotiations, and Collaborations of Regional Integration: A Perspective on Sustainable Management of Water Resources in Pingtung Plain, Taiwan

**Shuchen Tsai** [1] , **Suhsin Lee** [2] , **Zhe Zou** [1] **and Tajen Chu** [3],*

[1]  College of Arts and Design, Jimei University, Xiamen 361021, China; tsaishuchen@jmu.edu.cn (S.T.); zouzhe100@126.com (Z.Z.)
[2]  Department of Geography, National Taiwan Normal University, Taipei 10644, Taiwan; shlee@ntnu.edu.tw
[3]  Fisheries College, Jimei University, Xiamen 361021, China
*   Correspondence: chutajen@gmail.com

**Abstract:** A consultative and cooperative perspective on water management is vital in regional sustainability. However, previous approaches often failed to consider the complex trade-offs involved in water resource allocation. This study explores theoretical perspectives on regional integration as a policy goal through the process of water allocation. The main purpose is to explain new areas created by allocation and regional integration with local-scale cases. The connections between post-structuralism, fragmentation, and heterogeneity are explored with five case studies of groundwater zoning: a Xinyuan buried pipe compensation, a Ligang well closure and power shutdown, a Dachaozhou artificial lake, a Wandan hold back well, and a Meinong anti-deep-water wells. Along with the case studies, secondary literature, interviews, and a field investigation were used. The time span of water conflicts was from 1973 to 2019, and the spatial scope is the groundwater distribution area. The study found that regional integration and dynamic balance are each other's subject and object, which was empirically verified through the water conflicts in agriculture and the semiconductor industry. Regional integration comes through scaled and flexible methods of operation, and is produced through special market agents and post-structural spaces. In the process, the imbalance caused by conflict must also be adjusted and evolved through cooperation, competition, and negotiation to maintain the dynamic balance. This involves internal treatment of the local government, external integration of the central government, and technological evolution within the organization. Accordingly, several suggestions are proposed that may be helpful for sustainable water resource governance. In summary, this study makes up for the shortcomings of water management patterns constructed by simple spatial overlaying of regional integration. Our findings could effectively enhance negotiations and collaboration in water management for regional sustainability.

**Keywords:** sustainability; regional integration; heterogeneity; adaptation and evolution

## 1. Introduction

Water management is a complex affair involving many competitive negotiation and other uncertainties. The complexity increases when the goal is to combine the multiple benefits of often competing water uses, such as hydropower, industry, livelihoods, and irrigation, while reducing natural risks (flood control) and meeting environmental requirements [1]. Especially when water management covers multiple water spaces, it often causes conflicts between authorities or organizations with different interests [2,3]. Many issues can lead to conflicts, including waste disposal, granting of licenses, restrictions on use [4], and violation of agreed conditions [5]. When shortages or droughts are present, conflicts of course tend to become more critical [3,4]. Previous studies have examined water conflicts in large-scale, continental, and arid regions, and regions with increasing population, but

there is a lack of studies on small-scale, island-type, humid, and high rainfall regions, and regions with population decrease.

Regional integration is the process by which neighboring countries or governments enter into agreements to cooperate. Objectives can range from economic to political to environmental, where commercial interests are the focus of achieving socio-political and security goals [6,7]. Integration helps two or more countries overcome divisions that impede the flow of goods, services, capital, people and ideas [8]. Specifically, regional integration requires that cooperation between bilateral and multi-party national or local governments can be facilitated through common physical and institutional infrastructure [9]. De Lombaerde and Van Langenhove [10] describe it as a territorial phenomenon at the national level and create new forms of organization. Some scholars regard it simply as the process of strengthening interaction between countries in a specific region in terms of economic, security, political or social and cultural issues [11,12]. In short, regional integration is to combine various units in a region into a larger whole when faced with many difficulties, focusing on environmental regulation and integration. Therefore, the use of water resources often causes international conflicts, especially in transboundary rivers. How this conflict is negotiated will become more important to bilateral countries as it is used more and competition emerges [13]. In this regard, this is not only possible but beneficial, whether through formal organization or effective liaison, in order to avoid conflict [13].

*1.1. Background of Taiwan's Water Conflicts*

In Taiwan, the average annual rainfall exceeds 2500 mm and the average relative humidity is 70–80%; it is also an island with abundant water resources [14,15]. Due to geographical factors such as steep terrain, the river water flows rapidly into the sea. Therefore, the rainfall time is uneven and mostly concentrated in the wet season, which makes the utilization and management of water resources very difficult [16].

Since 2000, when Taiwan joined the WTO, agriculture declined [17,18] and industrial water demand increased. The government's method of water allocation was to encourage the use of fallow fields and allow water to be allocated from agriculture to industry [19,20], especially to leading industries such as the semiconductor industry, which was the key to driving and promoting Taiwan's position in the global industrial chain. In 2002, Taiwan proposed a two-trillion two-star policy to support the semiconductor and image display industries. This policy allowed the country to monopolize technology and gain a strategic position in the international economy [21,22]. In recent years, the South Taiwan Semiconductor S Corridor has been promoted; it is a continuation of the two-trillion two-star policy, with the intention of forming a cluster of semiconductor supply chains in the Southern Science Park [23]. Because wafer fabrication and packaging consume a large amount of ultrapure water, according to statistics on the production of 8-inch wafers from 15 wafer factories in Taiwan, for an average of 30,000 wafers per month, a total of 208 $m^3$/h of ultrapure water will be consumed. For the processing capacity of a factory producing 12-inch silicon wafers, the amount of pure water consumed will greatly increase to about 468 $m^3$/h, which is a great burden on water consumption and cost [24–26]. Most water consumption problems focus on agricultural products [27–30], while the discussion of water consumption in the semiconductor industry is relatively lacking. However, the semiconductor industry has been the biggest beneficiary of the global economy in recent years. It should be examined on the issue of water environment sustainability. Relevant studies also pointed out that business organizations can be held accountable [31]. Research also shows that the globalized economy is one of the main causes of water conflicts [3,32–34]. The continuous rise in water demand in this industry also means that pressure on the water supply is also increasing [35–37]. The main geographical location of the S corridor is Tainan City and Kaohsiung City, and the water supply comes from Pingtung Plain in the south. This study will show that the southward migration of Taiwan's semiconductor industry is the main reason for the government to formulate regional integration policies for water environment, and the water conflict case will prove the direct correlation between the two.

However, river pollution, declining groundwater level, uneven rainfall, and typhoons all threaten the stability of the water source on the plain. If there is no sustainable plan to address the problem, water shortages and conflicts will continue to occur.

Water allocation according to the national development goals must focus on regional stability, so that there will be sustainable long-term benefits [38]. Therefore, the Taiwanese government's latest water environment construction policy aims to "promote the country's economic transformation, balanced development and regional integration" to solve the water distribution battle and achieve environmental sustainability.

### 1.2. Purposes of This Study

The purposes of this study are as follows: (1) to use local-scale conflict cases to explain methods, roles, and spaces in the process of regional integration; (2) to discuss the spatial development of regional integration from the perspective of post-structural theory; and (3) to explain the adaptation methods used to maintain the sustainability of the water environment in the context of water conflicts.

## 2. Materials and Methods

### 2.1. Study Framework and Design

Research methods including case studies, secondary literature analysis, fieldwork, and in-depth interviews were used. This study focused on five cases in the groundwater area of the Pingtung Plain, covering the period 1970–2019.

According to the collected data, although the study cases have a common theme, they do not intersect in time and space. In order to achieve the research objectives, the following strategies were proposed:

(1) Here we take an underground rhizome as the research strategy to connect the heterogeneous characteristics of the case, such as the supplementary location, event time, and social relationship, and then pick up after there are no more data or broken data to connect other characteristics, such as the cases of Xinyuan and Lingang in the same period. Based on the principle of heterogeneity, the image of an underground rhizome is presented, by which regional integration (how), role-playing (who), and space formation (where) are identified by this method.

(2) Explain the new regions generated by regional integration from a theoretical point of view. The combination of heterogeneous space is the focus of post-structural geography. Therefore, the new region generated by water distribution uses the Concept of Deleuze with regard to territorialization, deterritorialization, and reterritorialization. They refer to investing energy in specific areas, withdrawal of energy, and reinvesting energy elsewhere [39]—in other words, which actions linking space to political economy and Deleuze's conception was focusing on a two-way or multi-directional mutual subject and object [40]. For example, water and silicon are inorganic and organic substances, respectively. The purity of water is invested to make silicon wafers—territorialization. After the withdrawal of pure energy, it becomes sewage and chips—deterritorialization. sewage purified into reclaimed water invested to industrial zone—reterritorialization. On the other hand, the chips and circuits being invested in the global economic market is also reterritorialization.

(3) Find a practical scheme to maintain the balance of the water environment in the new area generated by regional integration.

This study adopts the research approach of geographical political economy. From the point of view of economics, water is a factor of production, not just a natural resource, especially in the semiconductor industry [26]. When water distribution becomes capital, it can be put into production and generate income distribution. However, from a political point of view, there must be a good allocation model for water as a natural resource in order to achieve governance benefits. Therefore, in this study, the term "allocation" or "distribution" will be used depending on the context of the article.

The case of water conflict in Pingtung Plain is studied through poststructural theory in an attempt to explain the policy intent of regional integration through theory and experience, and to seek solutions to the problem of human–land conflict caused by insufficient water-spatial structure.

### 2.2. Study Area

The Pingtung Plain is located at the southwestern tip of Taiwan. Because the plain is located between a fault zone and a trench, it forms a complete geographical area Figure 1 [41–43]. The plain is composed of two topographic areas, an alluvial fan and an alluvial plain, caused by major waterways: Gaoping Stream, Donggang River, and Linbian Stream [44]. The area of the plain below 100 m above sea level is about 1130 km². It ranges about 50 km long from north to south, and about 20 km wide from east to west. The natural range of groundwater zoning is formed by the boundaries of the foothill belt, the alluvial fan of the river, and the coast [45]. Because the alluvial fan is composed of gravel, mud, sand, etc., the aquifer is deep and permeable. Together with the main rivers, the whole plain area is a huge water storage facility. The total area of the groundwater region in Pingtung Plain is about 1236 km², as shown in Figure 1.

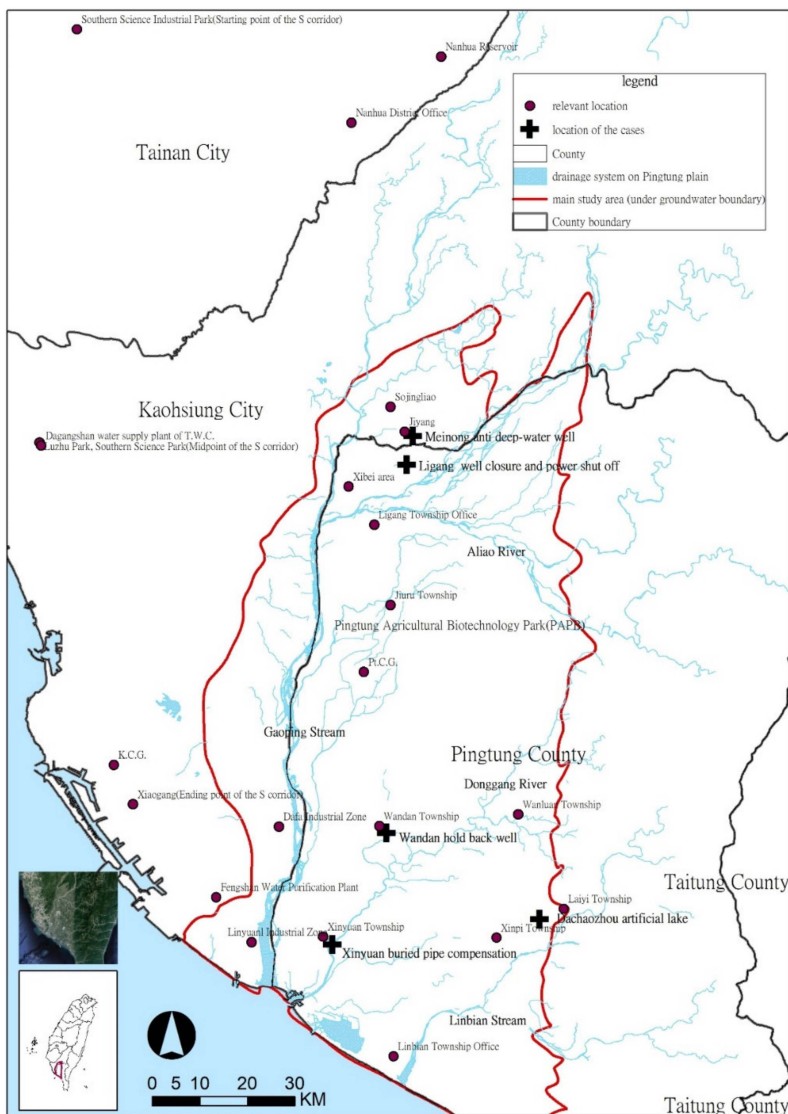

**Figure 1.** Study area, Pingtung Plain, Taiwan. (Source: Groundwater zoning boundary data cited from [46]).

*2.3. Data Collection and Information Sources*

There were two main sources of data. The first sources on water allocation include official and institutional information, official statements, internal meeting minutes, research reports, planning reports, survey reports, etc. Historical water conflict events and data of management organizations were collected. Sources included newspapers, magazines, and social media, as well as various planning reports and statistical data from the Ministry of Economic Affairs (MEA) and the Taiwan Water Company (TWC) and its related sub-organizations. In addition, announcement information, planning reports, and meeting records were collected from the Pingtung County government. The second data source was interview records from various agencies, representing unofficial opinions and descriptions. After the research sources were collected, EXCEL was used for coding. Then, the distribution data figure of events and time were obtained from which to judge the concentration time of water conflict events, and finally the time range of the research was determined. Total 133 water allocation incidents, 82 water shortage incidents, 37 tap water incidents, and 44 water pollution incidents Figure 2.

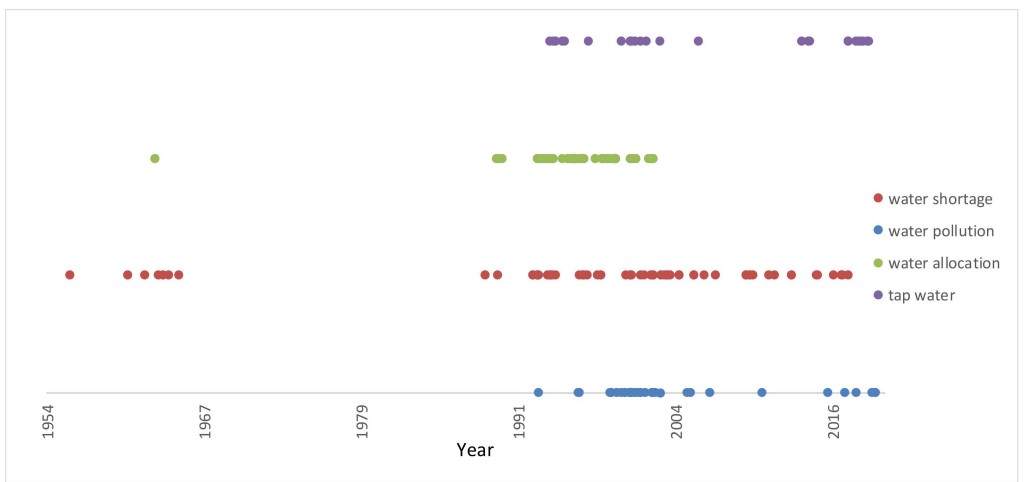

**Figure 2.** Number and time of water events (1960–2019).

The distribution data of events and time shows that water conflict events began after the 1960s and increased sharply in the 1990s. Therefore, the water conflict events that occurred between 1960s–1990s will be indicative.

*2.4. Case Studies*

Five cases were used to explore water distribution, water scarcity, conflict, and negotiation (Table 1). Objects included residents, farmers, religious groups, organizations, and county governments. A network of nodes and connections was constructed, including attitudes, experiences, knowledge, and awareness. These contents show the experience of water shortage in life and industry, ordinary people's knowledge of the local hydrological environment, the formation of environmental risk awareness, mobilization and gathering on the spot after the news of water distribution, and calls to expand the scale. Although the five cases share common themes, they do not overlap in time and space.

The first and second cases are located in the north and south of the Pingtung Plain, but their time of occurrence is similar. They involved compensation for buried pipes in Xinyuan Township and power shut-off for closed wells in Ligang Township. Although they are located in different watersheds, they are both cases of compensation due to cross-regional distribution in Pingtung Township. The geography spans two counties and cities along Gaoping Stream, and the time range is 1975 to 1994, which makes them indicator cases of water conflict. The third case is Daochaozhou artificial lake, which is the first and largest artificial water intake facility in Taiwan. It is located along the upper reaches of

Linbian Stream. Construction of the first phase started in 1999 and ended in 2013. Wandan Township represents the incidence of water conflict between farmers and the government from 2011 to 2014. The fourth and fifth cases are anti-deep-water wells in Meinong District, which are still under negotiation.

**Table 1.** Basic descriptions of five study cases.

| Case | Location | Related Parties | Claim | Situation |
|---|---|---|---|---|
| Xinyuan Township buried pipe compensation | Xinyuan Township, Pingtung County | 1. Xinyuan villagers 2. Xinyuan villagers' representative 3. Pingtung County Government (PtCG) | Compensation | Complete pipe burying and monetary compensation |
| Ligang Township, well closure and power failure | Ligang Township, Pingtung County | 1. Ligang villagers' representative association 2. Pingtung County Government | Well closure and compensation | Complete well closure and monetary compensation |
| Daochaozhou artificial lake | Xinpi Township, Pingtung County | 1. 185 County Road Self-Rescue Association 2. Pingtung Environmental Protection Alliance | Risk assessment/compensation/safety improvement and phase II EIA | Artificial lake is under construction and traffic safety is being improved |
| Wandan Township hold back well | Wandan Township, Pingtung County | 1. Wandan Township hold back well self-rescue association 2. Wandan villagers' representative association | Stop | Well-sinking project is completed and members of self-rescue association are subject to judicial determination |
| Meinong District anti-deep-water wells | Meinong District Kaohsiung City | 1. Anti-deep-water wells self-rescue association 2. Chishang and Meinong villagers | Stop | Project is on hold |

### 2.5. Field Investigations and Interviews

Field investigations and interviews were conducted from 2014 to 2019. The main interviewees were local people familiar with the situation (school principals, members of NGOs, public opinion representatives, farmers, residents, and members of self-rescue associations), local officials (Taiwan Agricultural Research Institute, Seventh River Management Office, Ministry of Economic Affairs (water resource authority (WRA)), Water Resource Planning Institute of WRA, Water Resources Bureau of Kaohsiung City, Department of Urban and Rural Development of Pingtung County, Department of Agriculture of Pingtung County), and professionals (professor of hydrology). A total of 20 interviews were conducted, and the responses were snowballed to connect related objects.

*2.6. Concept of Regional Integration: Theory and Claims*

2.6.1. The Environmental Turn of Neoliberalism

In this study, the region is placed in the context of neoliberalism. The reason is that the post-Fordist growth mechanism began to seek adjustment at different geographic scales. Region is one of them; it can adjust the space and can operate more flexibly. The importance of scaling has been highlighted in the development of the world economy [47,48].

One regional scaling method was to expand to the natural environment, forming a neoliberal environmental turn [49,50]. In this period, the government began to intervene in economic and political thinking [51–53], because economic efficiency and public interest were equated [54]. Therefore, the government began to establish administrative efficiency based on the economy. The main economic models include outsourcing, deregulation, privatization, and commercialization [55]. In this context, it can be said that neoliberalization loosened the boundary between the state and the market, integrating the two heterogeneous roles. Castree [56] proposed the concept of environmental fixes and regarded neoliberalism as a measure of a set of action plans. At the same time, the author put forward incisive definitions, including privatization, marketization, deregulation, reregulation, market agency of the quasi-public sector, and the construction of flanking mechanisms in civil society [57,58]. This definition is an important basis for this study to analyze the regional integration agents, and it is found that the market agent is the key role in the implementation of regional integration. According to Lane [59], New Public Management (NPM) was proposed to describe how the management technology of the private sector is applied to the public service, and this study uses the viewpoint of neoliberalization to explore the public service tasks of the public sector through the role of quasi-official market agency, emphasizing that the quasi-official mediates between the official and the private sector with multiple roles to achieve the purpose of regional integration.

2.6.2. Post-Structural Theory

The book *Anti-Oedipus: Capitalism and Schizophrenia* [39] *Capitalisme et Schizophrenie. 2 Mille Plateaux* [40] was used to explain the spatial development of regional integration. The creative discourse emphasizes cross-boundary thinking and multiple experiences, allowing us to understand the fragmented world. The "connections" of fragments are understood by thinking about deterritorialization and reterritorialization; that is, by constructing the context, which can reveal important meanings, including the decisive factors and the basis of power relations. This allows us to create new concepts, perspectives, and theories to address problems.

The intersection of new regionalism and neoliberalism will reopen and connect the boundaries in the area. This multiple and complex relationship resembles the territorialization, deterritorialization, and reterritorialization of nomadic space [40]. Deleuze emphasizes cross-boundary thinking with multiple experiences and experiences, so that thoughts can be deterritorialized and reterritorialized. The book *A Thousand Plateaus* takes wasps and orchids as an example of mutual deterritorialization and reterritorialization. Wasps are part of orchids' reproductive organs and were thus an instance of deterritorialization, by spreading orchid pollen to make orchids reterritorial. Both wasps and orchids are fragments; two completely heterogeneous fragments are interconnected, continuously connected, and control each other. The heterogeneity forms underground rhizomes, which promotes thought deterritorialization and develops spatial thinking [60].

**3. Results**

*3.1. Regional Integration with Flexible Scale*

3.1.1. Xinyuan Buried Pipe Compensation

From 1973 to 1979, the government promoted 10 major construction projects, including China Steel Corporation, China Shipbuilding Corporation, and China Petroleum Corporation, which settled in Kaohsiung. These are all industries that require water. In 1976, in order to transport water from the Donggang River to Kaohsiung, a river weir was built in

Xinyuan Township, Pingtung County. This is an important facility for the construction of water-requiring industries. However, this structure resulted in environmental degradation, including subsidence, soil salinization, and seawater intrusion. Residents began to complain to the county government, stating that the main reason was the water intake facility. Therefore, the PtCG made a request for compensation to the water company of the Taiwan provincial government. After the Pingtung College of Agriculture presented a disaster assessment report, both sides immediately negotiated the amount of compensation. In 1990, the county magistrate asked the province, the city, and the water company to provide the funds at the provincial and city chiefs' meeting, and sought funding to repair the rivers and improve water pollution. The Kaohsiung County Government reached a compensation agreement through negotiation. In cases like this, the water company of the Taiwan provincial government has full authority to deal with the county government and the township office for compensation. The lack of water could cause a great industrial crisis. Xinyuan Township and the PtCG have bargaining chips. It is also known that the water company has very high authority to deal with conflicts. This shows that the institution itself has highly flexible execution and operation capabilities Figure 3.

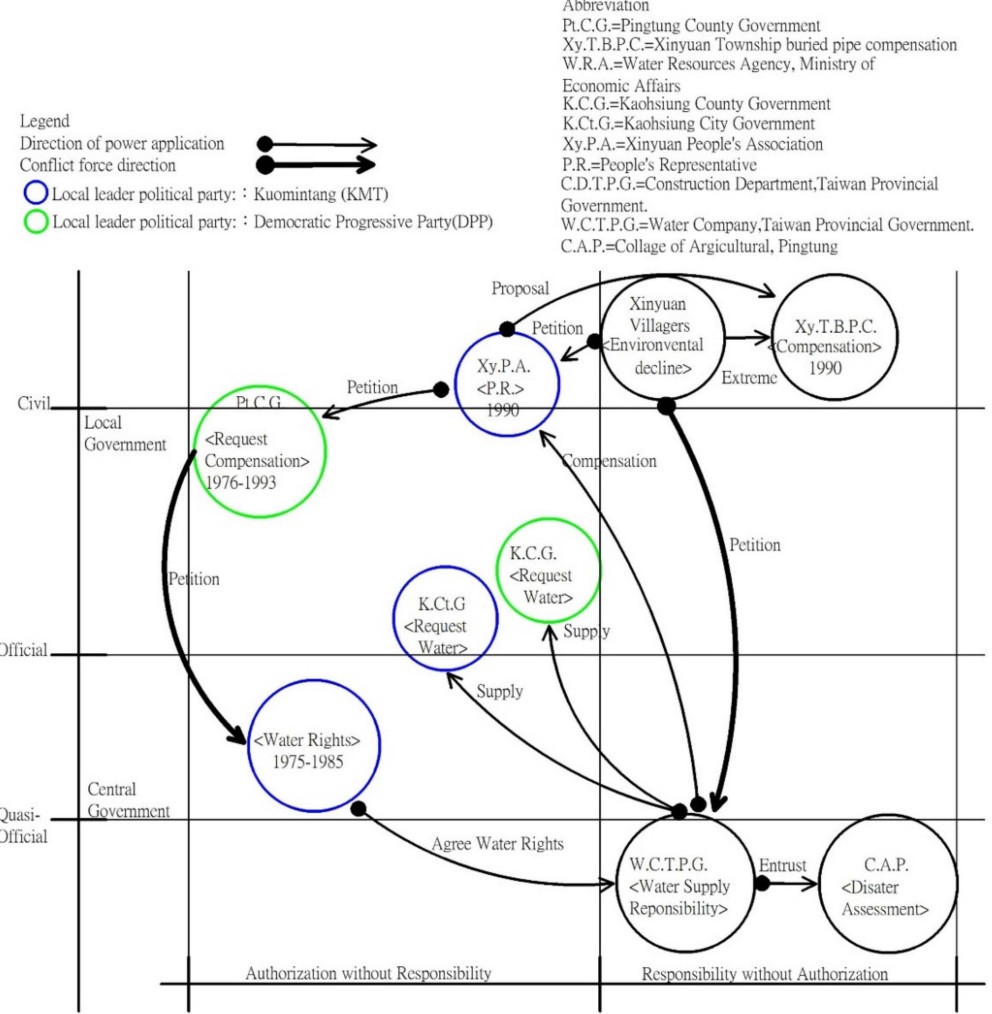

**Figure 3.** Flexible scale of Xinyuan.

## 3.1.2. Ligang Well Closure and Power Shut-Off

Since 1974, Kaohsiung City and Gangshan Township and have relied on 13 deep water wells in Ligang Township. They extract 127,700 cubic meters per day, supplying 420,000 people with domestic and industrial water [61,62]. In 1987, the water company

dug five deep wells in the Shoujinliao area of Kaohsiung County, which caused panic among residents in Ligang, and they jointly opposed it [61]. (In 1989, the Ligang Township Congress predicted that the groundwater level would drop, with accompanying reductions of crop yield and demand and the cessation of pumping.) The PtCG shut down the power on its own accord when the water rights of two wells expired, and the water company appealed to the Provincial Construction Department [62]. The PtCG implemented the water cut-off, and the Kaohsiung County Government responded by cutting off the road, preventing people from entering Kaohsiung [63–66]. In 1988, the Ligang Township Congress petitioned the PtCG and requested that the TWC carry out evaluation work. In 1989, the TWC agreed to conduct an evaluation and appraisal with Fengjia University, and in 1990 signed a consultation resolution. The conclusions include a deadline to stop pumping, funding subsidies and pipeline compensation, water conservancy facility subsidies, etc. The well closure and power shut-off incident in Ligang ended in 1994 after the well shut-off was completed.

In this case, we can see that the TWC faced a major crisis with the sudden cessation of water rights. Under the conditions of the provincial government's transfer of water rights, the flexibility of execution capacity was lost. After the Provincial Government Construction Agency devolved water rights to local governments in 1983, the water distribution relationship was extremely unstable. With water rights, Pingtung and Kaohsiung Counties joined forces to put Kaohsiung City's water supply below the political level, effectively reversing the unfavorable state. Ligang Township successfully resolved the crisis after a coopetition negotiation in 1990 through an effective political connection (Figure 4).

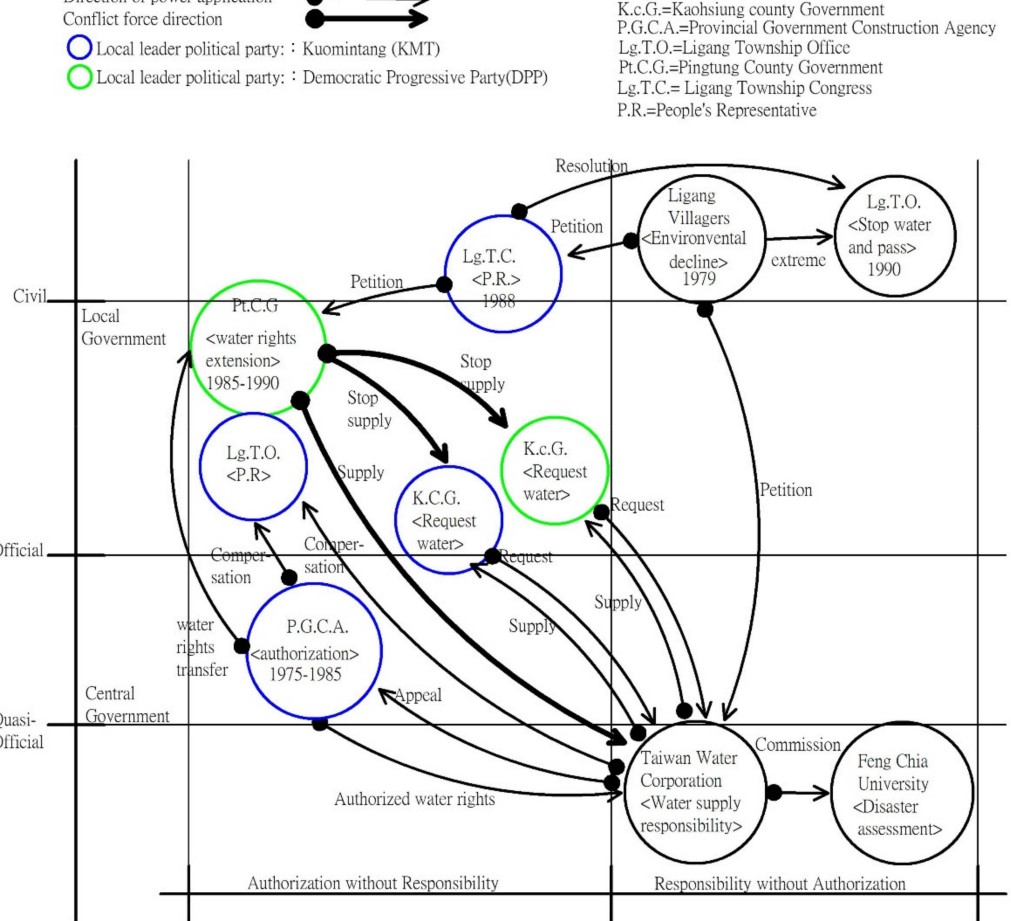

**Figure 4.** Flexible scale of Ligung.

### 3.1.3. Dachaozhou Artificial Lake

Dachaozhou artificial lake is an artificial groundwater recharge lake located at the junction of Laiyi and Xinpi Townships in Pingtung County. It is used to collect the excess water of Linbian Stream during flood periods, and makes use of the geological conditions in this area with good infiltration effect to refill the underground aquifer [67,68]. At the same time, it has the functions of flood diversion, flood control, and water conservation. In the water shortage crisis in 2021, the artificial lake played a great role. Before the project started, due to the opposition of local residents and environmental groups to the construction of Meinong and Majia Reservoirs, the Pingtung County Government was forced to promote the artificial lake project. However, during the construction, gravel trucks frequently caused traffic accidents with villagers in Laiyi Township. Therefore, the 185 County Road Self-Rescue Association came out to oppose the project. When there is conflict, the government and engineering companies keep the negative voices to a minimum. The exclusively local scale is the biggest feature of this case; power can therefore be exercised and conflicts can be resolved under the conditions of rights and responsibilities Figure 5.

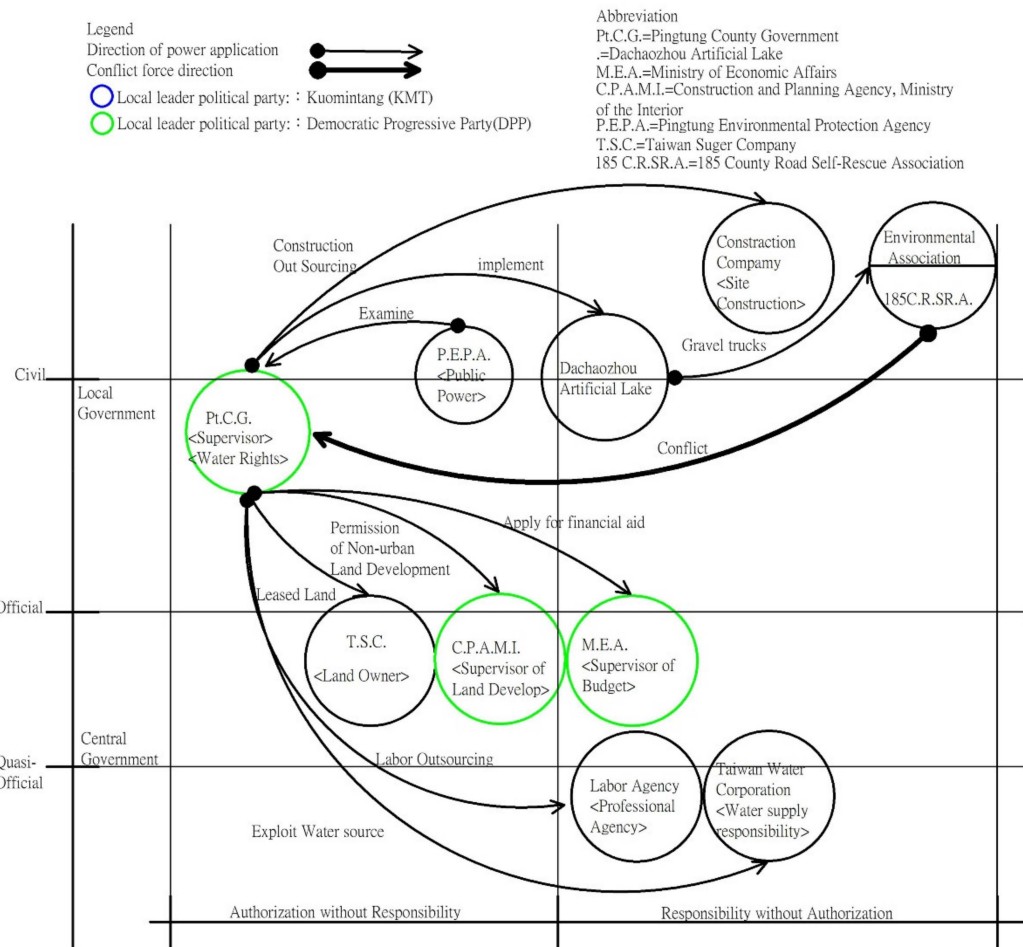

**Figure 5.** Flexible scale of Dachaozhou.

### 3.1.4. Wandan Hold Back Well

There is no running water in Wandan Township, Pingtung County, and residents in several villages have been infected with leptospirosis from drinking the groundwater. The county magistrate promised to install tap water in Wandan Township. In 2010, the application for water rights from the TWC was approved. The drilling of wells to extract groundwater can cause problems for farmers, such as a lack of water for irrigation and subsidence of the land. The corporation contracted to drill a well, but was blocked by the

protests of the villagers; as a result, the project could not be constructed and the contract was terminated. The purpose of the hold back well is to stop the subsidence of the land.

In this case, it can be seen that power and responsibility are divided into four dimensions, creating an innovative executive portfolio. Each unit fulfils its responsibilities, which it is allowed to "administer according to law", and the local people cannot change the way of governance Figure 6.

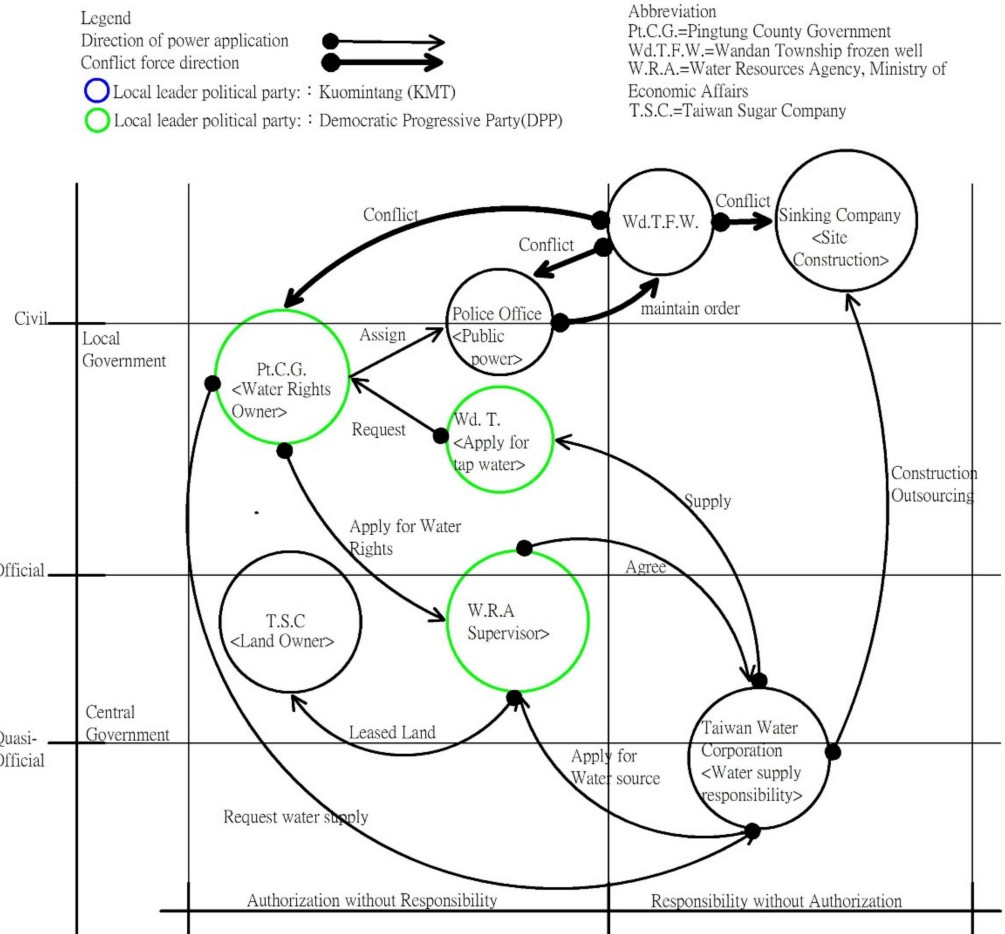

**Figure 6.** Flexible scale of Wandan.

### 3.1.5. Meinong Anti-Deep-Water Wells

In the case of Meinong, the conflict directly involves the water corporation on the front line. Based on the support of residents and non-profit organizations, local and central representatives have the same opinion. They also formed a self-rescue association that can be expanded. Since the Ligung incident in 1979, Meinong's anti-deep-water wells have engendered a sense of opposition. In the local area, water conflicts forced the suspension of the strong water distribution policy Figure 7.

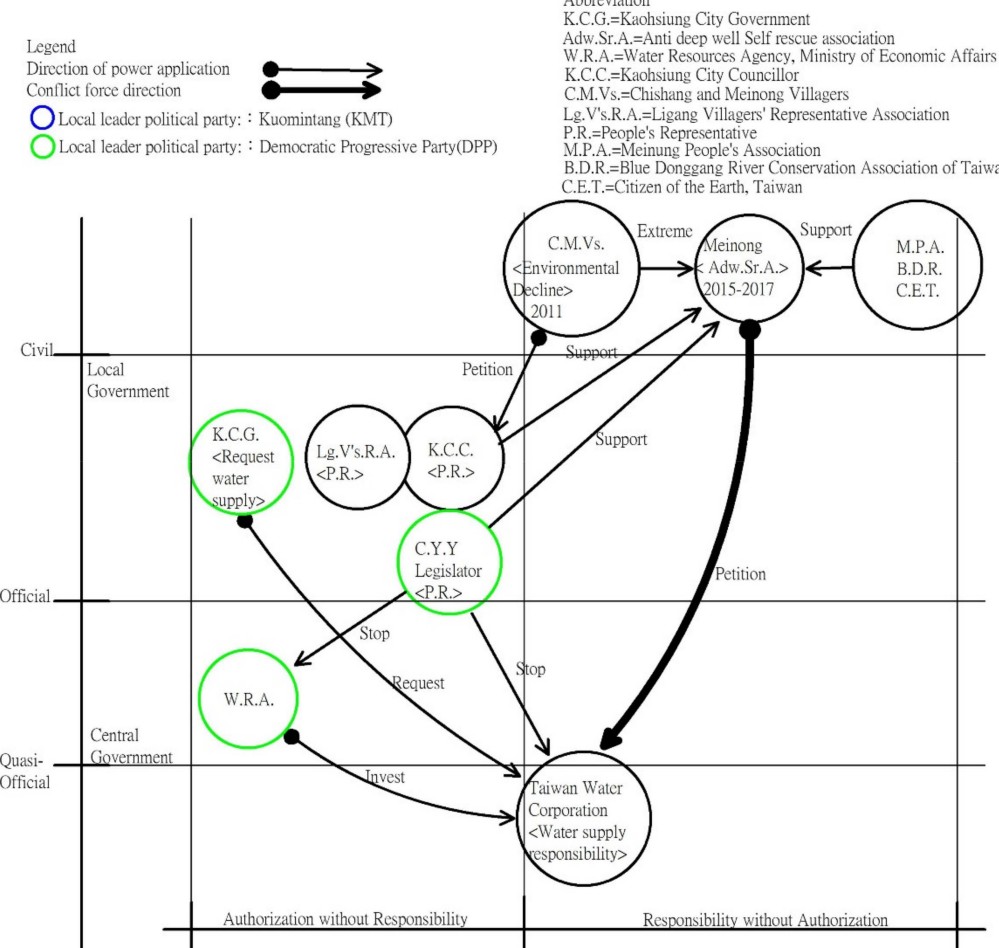

**Figure 7.** Flexible scale of Meinong.

*3.2. Regional Integration: Multiple Authorities and Responsibilities (How), Market Agency (Who), and Post-Structure Space (Where)*

The authority and responsibility relationship regarding water distribution is shown for the five case study conflict events. The contents include water demand requirements, water rights issuance, protest handling, on-site construction, appraisal and evaluation, monetary compensation, construction compensation, agreement contracts, etc. Various agencies or institutions use different contracts or entrusted relationships, showing the stakeholders involved at different levels. For example, the WRA entrusts the TWC to carry out the water intake project; the TWC outsources the well-sinking and pipe-laying work to private companies; after the completion of the water intake project, the WRA transfers it to the TWC for operation and management; the TWC leases land from Taiwan Sugar Corporation; the township office applies the key points of the work of good-neighborliness formulated by the TWC; the township office and the TWC negotiate a compensation contract.

However, once disputes or conflicts occur, they are restrained by the contractual relationship, resulting in structural problems. This is qualified by the multiple authority and responsibility scale (MARS), which categorizes relationships into their scope according to responsibilities and roles. Then line segments are used to connect relative relationships and clearly present all power distributions. This study opens up a parallel space–time concept, placing the five cases on the same platform for discussion. All parts of the case are spread out, such as dismantling, classification, and reorganization, to restore the role-playing within the political and economic mechanism.

From the authority and responsibility relationship diagrams of the five cases, it can be seen that the MARS in the decision-making process will lead to distribution conflicts in

non-governmental and quasi-official locations. The main reason is that the TWC, which has front-line personnel at the scene, directly faces the masses, but because of the division of responsibilities and authority, they are often ignored in the process of power analysis. However, the characteristics of its multiple components are the necessary abilities of people with multiple levels of authority and responsibility. The following is an individual analysis of the distribution of multiple authority and responsibilities in the five cases. The interactions of all the characters in these events were attempted to be understood, while observing the multiple characters of TWC at the same time using scale when dealing with conflict (detailed Figures 3–7).

*3.3. Marketing Agency*

When dividing the public sector and the market in the MARS, the TWC plays a vital role for stakeholders. It uses the characteristics of execution, operation, and cooperation in the five cases to flexibly handle all conflicts.

According to the defining features of neoliberalization [57], the water company is a quasi-public enterprise that has the responsibility of allocating water. Corporate social responsibility (CSR) should tend to become important for sustainable water management [69]. Politically, the civil service regulations apply. Economically, it is a for-profit business entity; it therefore has the characteristics of execution, management, and cooperation [70]. Its main role is to cooperate with and carry out the tasks assigned by the WRA. However, it is also an independent for-profit business entity with a monopoly in the market providing people's daily necessities. As the Ministry of Economy supports the development of new water sources, it potentially has a niche as an efficient and competitive market agent within the public sector system. Under the leadership of the state, it can perform continuous public service. On the other hand, it has become an efficient and competitive enterprise similar to private sector businesses. Through this CSR process, companies can communicate their current strategies, especially their sustainability practices [69]. Moral responsibility is especially needed in areas with weak governance or limited water supplies [69,71]. According to Sánchez-Hernández and Robina-Ramírez [69], it is increasingly necessary for organizations and public institutions to be transparent and ethical, comply with existing legislation, and respect and meet stakeholder needs. Social responsibility becomes a strategic choice in order to improve the relationship between institutions and the citizens they serve [72]. This is why the TWC has always played the most important role in regional integration.

In Xinyuan and Ligang, the TWC acted as the executor to quickly handle compensation, whether it was burying pipes or closing wells, and maintained contracts with the local governments. In the case of Chaozhou artificial lake, it played the role of a co-operator, and the subsequent water supply part was transformed into the role of an operator. In the case of Wandan, it faced the masses as an executor and completed its tasks with the support of the local government. In the case of Meinong, it became a complete co-operator, completely obeying the instructions of the WRA. All in all, in the five cases, regardless of whether the water supply responsibility was fulfilled, the TWC could skillfully use its multiple characteristics to mediate between various roles, the public sector and the market, and complete its own tasks.

In the water distribution cycle, the TWC makes water measurable. In addition to the currency collected from water bills, it uses contracts to bind water space to compliant people, then convert it into a single currency, such as compensation fee, neighborly fee, tap water fee, etc. The currency is allocated to the local government for redistribution. The TWC invests in its own productivity and power with money and contracts and uses administrative power to continuously expand the amount and space of water supply. It allocates more water to the most resource-efficient industries, providing resources for economic survival. The resources are converted into currency and re-circulated in the capital cycle, supporting and consolidating each other to complete regional integration.

Although the TWC is a for-profit enterprise, its shareholders are all government agencies. It has the ability to convert water into currency, and has the credibility of

contracts. In the process of reorganization, local governments, as shareholders of profit-making enterprises, also play the triple roles of execution, operation, and cooperation. In the process of conflict, contracts made through coordination and collegiality reflect the essence of the existence of the state along with the currency. The contract presents itself as an instrument of subjectivization, which grants rights while imposing discipline. Conflict can be subordinated to political ends, and conquest is the result of the process. At this time, the TWC has become the dominated as well as the dominant party [40]. For example, in the case of Xinyuan, the compensation agreement shows the complete subjectivity of the state. Rights and discipline co-exist. During the back-and-forth negotiation of compensation fees and land plans over the years, the dual roles of the Kaohsiung Industrial Zone and the Xinyuan Pumping Station as the dominant and dominated were fully demonstrated. For example, in the case of Ligang, once the contractual relationship of water intake is established, both sides are the objects of exchange of creditor's rights. One hand supplies water and one hand gives money. When the water supply ends, the contractual relationship, with monetary compensation, also ends.

Market agents use contracts and currency to create territorialization. For example, two administrative regions on both sides of the Gaoping River were integrated into one water use area due to water allocation. Once the contract stopped due to environmental decline, the two administrative regions also deterritorialized. After deterritorialization, market agents must find targets that can meet the demand for water consumption to maintain regional stability. In order to pursue sustainable development, the places where water resources are exhausted are reconstituted to nature by means of competition, consultation or cooperation, so as to meet the stable living conditions. From the perspective of regional integration, it is a kind of dynamic balance Figure 8. Intersubjectivity of territorialization and deterritorialization is formed between opening and closing. After capital uses neoliberalism to get rid of state control, it forms a supranational deterritorial force with a transnational and global organization. Taking currency, the most direct quantitative form of capital, it acts directly in places and becomes the most deterritorializing element [40,73].

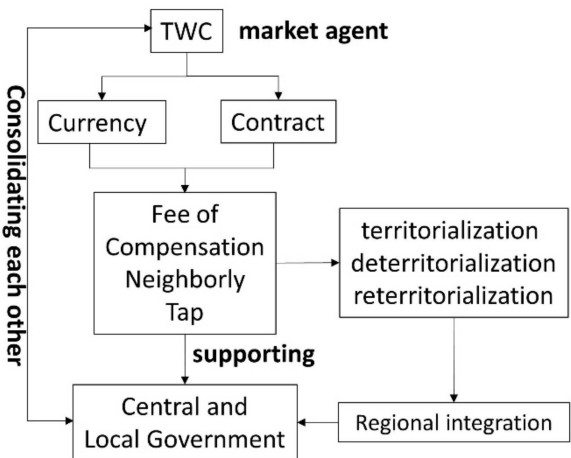

**Figure 8.** Diagram of market agent mechanism.

### 3.4. Conflict and Integration of Smooth Space and Stratified Space

The region before territorialization is a fuzzy and imprecise space, but the identification of space in the process of conflict manifests its existence. Until the data are entered and quantized, it exists as a stratified space. All conflict occurs during this transformation. The main reason for the conflict between distribution and regional integration comes from two spaces that cannot be put into dialogue: smooth space and stratified space. The conflict phenomenon of regional integration involves dialogue failure from post-structural space. The language used by farmers in smooth space comes from culture, life experience, industry, etc. However, in stratified space the focus is on the amount of replenishment, pumping,

storage, compensation, etc. The difference between analog language and digital language is the main source of conflict. Therefore, when digital capital is regionally integrated, currency with both real and virtual functions will be used as a common language.

From the two cases of Xinyuan and Ligang, it can be seen that when the residents complained about the declining groundwater level, the reduced agricultural production, and the subsidence of the land, there was no solution. After an investigation by a third-party academic unit, the specific quantitative data became the common language of communication between the two parties, and a consensus on a solution could be reached. In the Chaozhou artificial lake and Wandan, the self-rescue association and the county government are located in two spaces. They have no common language for communication. The main reason is that when the water intake project was not completed, there was no quantitative data to prove it. In Meinong, the self-rescue association proposed to prove that water from the deep well would be supplied for industrial use. Clear information becomes a bridge for dialogue, and supports a smooth space [40] that spans ethnic groups, regions, and currencies. A non-metric, centerless, rhizome-like multibody has properties but no walls or fences. Only in the process of reterritorialization does an identifiable form emerge. In the process of reversing the deep well, Meinong used eddy currents, starting from Jiyang at the center, and sent a message to the Ligung and Chisang areas (so-called Xibei area) to use street sweeping, virtual and physical gatherings, and collectively appeal to the power center; to fight against lobbying with knowledge and materials, and reject compensation and exchange; and to use every means to organize social cohesion. This action shows the process of deterritorialization and reterritorialization of nomads in the Xibei area.

The five cases generate new regions through the process of regional integration of territorialization, deterritorialization, and reterritorialization. They also present the characteristics of interconnection and mutual control of heterogeneous elements, forming underground rhizomes, with the transformation of water distribution into new regions, and the transformation of new regions into water distribution. The generation of this relationship, as stated by scholars, "ensures the deterritorialization of one side and the reterritorialization of the other side" [40,73].

### 3.5. Negotiation, Cooperation, and Competition of Regional Integration

The fluidity of water makes local spaces affected by regional integration policies. It also leads to open competition and cooperation to move toward the possibility of environmental sustainability [3,74–76]. In the study, five cases were found showing that conflicts in water distribution caused by economic development can often lead to the possibility of cooperation because of compensation. There are similar cases at the national scale; for example, countries along the Nile and Jordan River basins in the Middle East have developed a cooperative model based on the consideration of economic co-prosperity, despite continuous conflicts over water resources [13,74].

As another example, Israel and Palestine, in the bilateral cooperation governance of the Jordan River, learn from each other's experience and technical support. In 1999, the Nile River Basin countries formed the Nile River National Association. These countries coordinated the division of labor and technology, and initially achieved cooperation in the distribution of river water resources. There are also peace pipelines in Turkey, with buried pipes to export river water to eight countries including Syria, Jordan, and Saudi Arabia. South Africa's experience shows that water resource management can become a key factor in promoting economic development and regional integration. Politics will also focus on potential economic benefits to motivate competitiveness and productivity [13].

In Asia, the signing of a water supply agreement between Singapore and Malaysia in 1924 was the beginning of coordinating water resources between the two countries. Over time, new agreements have constantly been made. Over the years, from Gunong Pulai, Skudai River, and Johor River, we can see that Singapore has continued to expand its water demand and water intake. Although the water supply agreement between Singapore and Malaysia has continued [77], the water shortage crisis has also stimulated Singapore's

progress in desalination and wastewater treatment technology, and water autonomy before the end of the 2061 water supply agreement has been proposed.

In Taiwan, the semiconductor industry has a low tolerance to water shortages, and has stringent requirements for water quality and quantity. Therefore, technological changes have also been developed during Taiwan's water shortage crises. The Taiwan Semiconductor Manufacturing Co. (TSMC) strives for the maximum benefits by reducing water consumption and drainage losses, recycling wastewater, and increasing the water production rate, with a set goal of 20% of recycled water usage by 2030 [78].

Government departments have also proposed various solutions for regional integration on the plains, such as a groundwater replenishment program implemented on the Pingtung Plain [67]. In the middle reaches of the Linbian Stream, the Dachaozhou artificial lake was set up as a flood detention pond. Artificial lakes can be seen as a competitive solution to the geographic characteristics of heavy rainfall with rapid inflow to the sea and rapid infiltration, which, with a total area of 300 hectares, can introduce 5% of floodwater during heavy rains, acting as a safety valve.

In order to deal with the large amount of wastewater in the Pingtung Agricultural Biotechnology Park, an artificial wetland was set up in the center of the plain to purify the wastewater. Surface flow ecological construction was used first to divert the polluted water to the wetland, purify it, and then return it to Gaoping Stream, constructing wetlands with low cost, sustainable water conservation, and no secondary pollution. The purified water flows to urban rivers to improve the microclimate [79]. The government also requires the use of reclaimed water in newly established industrial parks [80].

## 4. Toward the Adaptation and Evolution of Regional Dynamic Balance

### 4.1. Cooperation from Local Governments: Internal Treatment

Xinyuan is an indicator case that is compensated for cross-boundary distribution. Since 1975, the water resources of the Donggang River have been continuously distributed to Kaohsiung. The water network connects Pingtung County and Kaohsiung City. The TWC transfers the water from the Gaoping Stream in Xinyuan to the Fengshan Water Purification Plant, and then deploys it to the Linyuan Petrochemistry Industrial Zone for use. In 2020, the water of the Donggang River in Pingtung County accounted for 51.75% of the total industrial water consumption in Kaohsiung City [81]. The water volume in the Donggang River is stable and high. Stable water volume is an essential requirement for industries. Therefore, the river is the key to maintaining regional economic stability [82,83]. According to a study of water resources in the Pingtung Plain [84], the alluvial fan of the Aliao River is diverted to the Donggang River. When the Linbian Stream fan is bounded by the topographical watershed, two-thirds of the area belongs to the Donggang River Basin. Therefore, the groundwater of either the Aliao Stream or the Linbian Stream fan flows into the Donggang River. As a result, the Donggang River has stable water resources to supply Kaohsiung.

However, the Donggang River has serious water pollution from animal husbandry [82]. It is the second most polluted river in Taiwan. The government needs to invest a huge amount every year to improve the raw water quality of the river [85]. In addition to building a water purification plant to improve the quality of raw water, since 2005 the PtCG has been actively trying to dilute the sewage by diverting purified water from the source. According to the project commissioned by the PtCG, the experimental data confirmed that the flow direction of the Linbian Stream refill water is from northeast to southwest [86]. During the flood season, Linbian Stream and Lili Creek are affected by lateral recharge, and the groundwater level flows to the southwest of Donggang River and Linbian Stream. It only flows south into the sea during the dry season [87]. Based on the characteristics of the river, the PtCG also chose to set up Daochaozhou artificial lake on the top of the alluvial fan of Linbian Stream to recharge the groundwater [67,68,88,89], because the stream's alluvial fan top has excellent infiltration. The stream water infiltrates quickly at the top of the fan to become underground water, which can quickly replenish the

groundwater [68]. Local officials have pointed out that "after the opening of the 300-hectare Chaozhou artificial lake, 100 to 200 million tons per year (at least 300,000 tons per day) will subsidize the Donggang River. The water infiltration conditions have been greatly changed, and 300,000 tons of water can be added" [90]. In a recent study on the water quality and quantity in the Donggang River, the results confirmed that the subsidized groundwater from Daochaozhou artificial lake has increased the water volume of the river and diluted the pollution of the stream, thus improving the water quality [91].

The coalescing alluvial fan in the Pingtung Plain is connected from north to south, forming one groundwater area. It can be seen that setting up an artificial lake from Linbian Stream to replenish the water volume of the Donggang River was a decision to maintain regional dynamic balance. The Ligang and Meinong areas in the upper reaches of Gaoping Stream belong to the same groundwater area, but because of the political divisions after 1920, they belong to different counties and cities. The political hierarchy across administrative regions has increased, and the governance method of recharging groundwater cannot be adopted. Dynamic balance must be maintained by negotiating regional competition at the central political level.

*4.2. Competition and Negotiation from the Center: External Integration*

In 2017, the deep-water well project was restarted in Meinong; mainly the Kaohsiung City Government actively encouraged the TSMC to set up a factory in Luzhu Park, Southern Science Park. The aim was to provide stable water resources as promised; also, in order to maintain the advantages of the semiconductor industry supply chain, the central government had to reopen regional integration and dynamic balance.

At the upper reaches of Gaoping Stream is the intersection of Tainan, Kaohsiung, and Pingtung. In 1992, the location of the Meinong Reservoir was quite controversial. At that time, it caused the most influential water environment movement in Taiwanese history, and also led to the formation of an important environmental sustainability group [92–94]. Unfortunately, their attempts ultimately failed. Since then, the central government has made internal organizational adjustments to cope with the next stage of regional integration. After 1996, the Reservoir Management Bureau began to be restructured into a Water Resources Bureau. In 2002, the existing water conservancy agencies were consolidated and the WRA was established, which is the highest administrative unit of water conservancy in the country. It can be seen that the institutional transformation showed the emphasis on water resource management.

The Jiyang artificial lake and Gaoping great lake proposed by the WRA are intended to address the daily 500,000 tons of long-distance water and industrial water shortages in Kaohsiung [95,96]. In other words, they are part of a new scheme to replace the function of Meinong Reservoir [97], but they ended at the EIA stage.

After these plans failed, the central government began to develop adaptation plans. At the same time, it was proposed to "rent the existing water wells from Taiwan Sugar Corporation. Using the old wells to improve and clean the wells to make up for the original plan, the two together can increase the daily water supply by 100,000 cubic meters" [98]. Based on five deep wells dug in Sojiliao, Kaohsiung County, in 1987 [61], the TWC initiated negotiation and compensation [98]. Finally, in 2011, 14 more deep-water wells were dug in the Sojiliao area. The daily water intake is 100,000 tons to fill the gaps in Kaohsiung's livelihood and industrial need for water, and maintain the dynamic balance of the current situation in the region by means of negotiated diffusion.

There is an intriguing paradox that explains the "not in my backyard" (NIMBY) attitude during facility selection. It means that the facility makes some residents feel that they do not want to be neighbors with it, which further means that the rich will not give up because of money, while those who cannot afford any losses are likely to sacrifice the environment for negligible compensation [99]. However, in the case of Meinong and Ligang, there were attempts to negotiate a water allocation plan after 2006 [100,101]. Even if the central budget was prepared, it was still unsuccessful [102,103]. Residents in the Xibei area

formed a social network across counties and cities and refused monetary compensation. This acted like a network of underground rhizomes strung together. Ultimately, the TWC deleted its budget and changed its allocation strategy [104]. This can also prove that even small-scale political bodies, such as local associations or communities [105], have the ability to propose water negotiations with the central government.

*4.3. Technological Evolution within the Organization*

The semiconductor industry has a high demand for water [26]. The evolution of water use developed during several water shortage crises in Taiwan. For Taiwan Semiconductor Manufacturing Co. (TSMC), the country's leading semiconductor producer and the world's largest contract chip maker, the production of a 6-inch wafer requires 1 ton of ultrapure water, an 8-inch wafer requires 3 tons of ultrapure water, and a 12-inch wafer is estimated to require 10–15 tons of ultrapure water [24,25]. The production of 8- to 12-inch wafers will increase the water demand by at least 67%. The company uses more than 150,000 tons per day. Between 2015 and 2019, TSMC's total water consumption increased by 70%, accounting for 3.4% of total industrial water usage.

In order to derive the maximum benefit from water, implemented measures include reducing water usage, recycling wastewater, and reducing drainage losses. In recycling wastewater, chemical techniques are employed to improve the quality for reuse. For example, the quality improvement is simplified through the use of a multi-pipe design, and the recycling system, which can include ammonium sulfate/developer, is completed to improve recycling efficiency and the utilization rate. The recycling system also uses activated carbon to remove oxidized substances, and then uses reverse osmosis (RO) membranes to remove impurities. The processes of recycling and reusing led to an annual wastewater recovery rate of up to 87.4% [106]. In addition, a target of 20% use of recycled water by 2030 was set in order to reduce the water demand of products by 24.7% compared with 2010 [78].

**5. Conclusions**

This study used water resources as the object to explain the subtle changes that occur with the fragmentation and heterogeneity of space in new regions and how to maintain spatial governance of the regional dynamic balance. The five conflicts in this study are five fragments. The main idea is focused on how to effectively explain the water conflict between fragments and regional integration. The underground rhizomes between the fragments reveal the regional integration: MARS (how), market agency (who) and smooth/stratified space (where). Language differences are the main source of conflict. The combination of real and virtual functions of currency represents a common language for regional integration. In particular, quasi-official market agents play a key role. Currency and contracts are used to invest energy in space and complete regional integration.

Some measures being suggested include local governments diverting water to replenish river water and dilute pollution and central governments submitting competing proposals for reservoirs, artificial lakes, and deep wells. These are part of an attempt to maintain the dynamic balance in the process of regional integration, in order to move toward the possibility of achieving water environment sustainability. Based on Taiwan's leading technological role in the semiconductor industry, the demand for ultrapure water has also driven the upgrading and development of water purification technology. Obviously, the demand and use of water is tighter and more difficult. In the future, related technologies can be transferred to public sewage purification, and industrial wastewater can be recycled into reclaimed water; islands can be used to develop seawater desalination technology and various water resources can be effectively circulated to achieve the goal of avoiding groundwater pumping. The process of integration is becoming more complex, negotiation and cooperation can serve as the principles of conflict and competition. The external network or platform and internal corporate responsibility and social ethics are the key to the solution. Our findings provide a spatial dimension to the consideration of

regional integration, and also take a big step towards the sustainable development of the water environment.

**Author Contributions:** Conceptualization, S.T. and S.L.; methodology, S.T. and T.C.; software, S.T.; validation, T.C. and Z.Z.; investigation, S.T. and S.L.; resources, S.T.; data curation, S.T.; writing—original draft preparation, S.T.; writing—review and editing, T.C.; visualization, S.T.; supervision, T.C. All authors have read and agreed to the published version of the manuscript.

**Funding:** This research was funded by an award from MOST for doctoral candidates in the humanities and social sciences to write doctoral dissertations, grant number 108-2420-H-003-006-DR, and by the 2021 Fujian Provincial Federation of Social Sciences Doctoral Support Project for research on Fujian's all-area tourism ecological planning from the perspective of rural revitalization (FJ2021BF047).

**Institutional Review Board Statement:** Not applicable.

**Informed Consent Statement:** Not applicable.

**Data Availability Statement:** Not applicable.

**Conflicts of Interest:** The authors declare no conflict of interest.

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
