# Peer review of "The Competitions, Negotiations, and Collaborations of Regional Integration: A Perspective on Sustainable Management of Water Resources in Pingtung Plain, Taiwan"

_sustainability, doi:10.3390/su14053040_

Round 1

Reviewer 1 Report

Thanks for the possibility to review this interesting article. I think the manuscript has the potential to be published with minor improvements. The introduction must include some concepts and ideas that are important in the article such as "regional integration" and "semiconductor industry" to be clearly linked to the purposes of the study described later. Conclusions have also to be improved to show the contribution of this paper to the knowledge on the field. Limitations and future lines of research must be also developed.

There are some errors to be corrected, expressions/concepts that need to be explained, and other minor suggestions for upgrading your paper:

  • Concept of Deleuze page 3 line 106
  • Post-structural Theory page 3
  • error in page 3 line 132
  • secondary literature (what do you want to express with "secondary"? what about "primarily"? Are you referring to information sources instead of literature? page 4 line 137
  • Does "Neo liberalization" in your paper refer to New Public Management (NPM)? Please explain and use the term NPM when needed
  • error in page 7 line 208 number "2"
  • Please rethink the term "new neoliberalism" on page 7 line 201 because new and neo are the same. What do you want to express with "new neo..."
  • error on page 9 line 272
  • page 13 line 356: I highly recommend introducing the term fourth sector, as you are referring to a for-profit business entity within the public sector system. Recommendation (not mandatory): Sánchez-Hernández, M. I., Carvalho, L., Rego, C., Lucas, M. R., & Noronha, A. (2021). The Fourth Sector: The Future of Business, for a Better Future. In Entrepreneurship in the Fourth Sector (pp. 7-22). Springer, Cham.
  • The concepts of "Re", "De", and "Re" territorialization must be defined and explained in the context of the article.
  • The majority of references in the article are technical papers. I highly recommend including more references from Academic Journals. For instance (not mandatory), reinforcing the importance of water management and the relationship with the sustainability of a business  [Sánchez-Hernández, M. I., Robina-Ramírez, R., & De Clercq, W. (2017). Water management reporting in the Agro-Food sector in South Africa. Water, 9(11), 830] or linking water management and social responsibility of local governments [Sanchez-Hernandez, M. I., Maldonado-Briegas, J. J., Sanguino, R., Barroso, A., & Barriuso, M. C. (2021). Users’ Perceptions of Local Public Water and Waste Services: A Case Study for Sustainable Development. Energies, 14(11), 3120.]

I hope my comments will help you to publish your paper.

Author Response

Thank you very much.

Reviewer 2 Report

My specific comments are below:

  1. Line 132 has some errors. Figure 1 was also not clear enough.
  2. Lin 272 also has error.
  3. There were some formats errors in reference.
  4. Please check the whole manuscript grammar and formats errors thoroughly.
  5. We hardly connect your content with paper title? May because you content about Negotiations, Collaborations, and Competitions of Regional Integration not enough?

Author Response

Thank you very much.

Reviewer 3 Report

The text is, in general, well-written and the exposed ideas transmit to readers a wisdom resulted from the past events and their analysis.
-line 46: "negative population growth" - let's not use euphemisms, simply say "population decrease" or something direct and similar.
-line 53: "In 2000" - most probably it is "since 2000".
-I was very curious what was the "two-trillion star policy" that the authors mentioned in the text; I found another translation in media: “Two-Trillion, Two-Star”, which makes more sense.
-the caption of figure 1: replace "scope of study" with "study area" or something similar, as there is exposed a territory, not an idea.
-the citation links of figures 1 and 4 are broken.
-line 172: "The first participant to be sentenced for a crime of public danger for a water conflict" - this phrase seems from a different story; please integrate it in the storyline with more details or remove it.
-line 502: "Cooperation from Local Governments: Internal Purification" - at a first glance, this subtitle sounds controversial. It is true that the text below it clarifies the meaning of "purification", but it is better to replace "purification" with "treatment" in the subtitle.
-line 531: "The stream water infiltrates quickly at the top of the fan to become submerged water, which can quickly replenish [...]" - "submerged" means under water! Replace "submerged" with "underground" or similar words. 
-through-out the document and in the abstract, the authors use "anti-deep wells" and "frozen wells" - these are not the best translations; these even suggest wrong meanings, such as wells filled with frozen water. Better replacements (just suggestions): anti deep-water wells, disabled wells?

Author Response

Thank you very much!

Reviewer 4 Report

The article is interesting, however, it needs improvement.

What statistical analyses were used to conduct the analyses. 
There is a missing graph showing what data was taken for analysis. 
Figures in my opinion are too complicated please rework the figures so they can. 
Please provide detailed numerical results obtained from the analyses. 
No discussion with publications describing problems from other areas of the world. 

Author Response

Thank you very much!!
